# Implementing Trauma-Informed Care—Settings, Definitions, Interventions, Measures, and Implementation across Settings: A Scoping Review

**DOI:** 10.3390/healthcare12090908

**Published:** 2024-04-27

**Authors:** Lene Lauge Berring, Tine Holm, Jens Peter Hansen, Christian Lie Delcomyn, Rikke Søndergaard, Jacob Hvidhjelm

**Affiliations:** 1Psychiatric Research Unit, Region Zealand Psychiatry, Faelledvej 6, 4200 Slagelse, Denmark; cdel@regionsjaelland.dk (C.L.D.); rso@regionsjaelland.dk (R.S.); 2Department of Regional Health Research, University of Southern Denmark, Campusvej 55, 5230 Odense, Denmark; jens.peter.hansen@rsyd.dk; 3Psychosis Research Unit, Aarhus University Hospital, Psychiatry, Palle Juul-Jensens Boulevard 175, 8200 Aarhus, Denmark; tinehol9@rm.dk; 4Retspsykiatrisk Forskningsenhed, Østre Hougvej 70, 5500 Middelfart, Denmark; 5Department for Forensic Psychiatry, Region Zealand Psychiatry, Faelledvej 6, 4200 Slagelse, Denmark; 6Clinical Mental Health and Nursing Research Unit, Mental Health Center Sct Hans, Copenhagen University Hospital—Mental Health Services CPH, 2400 Copenhagen, Denmark; jacob.hvidhjelm@regionh.dk

**Keywords:** implementation, mental health, multifaceted framework, organizations, scoping review, TIC, trauma, trauma-informed care

## Abstract

Traumatic experiences can have long-lasting negative effects on individuals, organizations, and societies. If trauma is not addressed, it can create unsafe cultures with constant arousal, untrusting relationships, and the use of coercive measures. Trauma-informed care (TIC) can play a central role in mitigating these negative consequences, but it is unknown how and in which way(s) TIC should be implemented. Our objective was to conduct a scoping review that systematically explored and mapped research conducted in this area and to identify existing knowledge about the implementation of TIC. The search was conducted on the CINAHL, Cochrane, Embase, ERIC, Medline, PsycINFO, and Web of Science databases, and more than 3000 empirical papers, published between 2000 and 2022, were identified. Following further screening, we included 157 papers in our review, which were mainly from the USA, Australia, New Zealand, and Canada, focusing on study settings, methodologies, and definitions of TIC, as well as the types of interventions and measures used. This review shows that TIC is a complex and multifaceted framework, with no overarching structure or clear theoretical underpinnings that can guide practical implementations. TIC has been defined and adapted in varied ways across different settings and populations, making it difficult to synthesize knowledge. A higher level of agreement on how to operationalize and implement TIC in international research could be important in order to better examine its impact and broaden the approach.

## 1. Introduction

Trauma may occur because of a harmful incident or series of events that are emotionally disturbing or life-threatening, such as violence, neglect, abuse, disaster, serious illness, or historical injustice. In the general population [1,2], most people experience at least one and, on average, three to four lifetime traumas, with rates of post-traumatic stress disorder (PTSD) ranging between 8 and 9% [3,4]. In psychiatric populations, the prevalence is much higher, and many (between 75 and 98%) report multiple traumas mentioned before [5]. Up to 50% of individuals with severe mental illness (SMI) also suffer from PTSD, but the actual numbers may be much higher, as trauma exposure and PTSD are generally underdiagnosed in individuals with SMI [6,7,8,9]. Traumatic experiences can have lasting negative effects and decrease the individual’s quality of life [10,11]. Severe trauma is also correlated with alcohol and drug use, homelessness, suicidality, self-inflicted harm, hostility, anger, lower levels of social functioning in SMI [5], and increased mortality [2]. In addition, a history of traumatic experiences may put individuals at risk of further traumatization, including harmful experiences within the psychiatric setting [12].

Individuals in helping professions, such as nurses, psychologists, and teachers, who play a central role in mitigating suffering among individuals with mental health problems, are also at risk of adverse outcomes, such as secondary traumatic stress or burnout in relation to their work [13]; for example, if they are exposed to verbal abuse or violence from service users or harassment from colleagues [14]. These reactions may be exacerbated if they themselves have experienced trauma in the past, such as interpersonal violence. If individuals suffering from unaddressed trauma are not helped, it can influence healthcare organizations in the short or long term. For example, it can create unsafe organizational cultures with constant arousal, untrusting relationships, frequent use of coercive measures, and struggles in coping with day-to-day life [11]. Thus, there is a significant need for programs or frameworks that aim to change how the helping professions work with trauma and prevent victimization.

One such approach is trauma-informed care (TIC), which is a strength-based framework for human services that assumes that individuals are more likely than not to have a history of trauma and acknowledges the role that trauma may play in the lives of service users, care providers, and the public. TIC was first described by Harris and Fallot in 2001 [15]. They envisioned how human services, such as hospital settings or schools, should commit themselves to providing services in a manner that welcomes and is appropriate for the needs of trauma survivors (p. 5) as, according to the authors, they construct a new meaning of themselves that is influenced by horrific events; the human care setting must respond with a trauma-informed approach, where they return a sense of control to the service user. TIC is holistic [15] and integrates six guiding principles: safety; trustworthiness and transparency; peer support; collaboration and mutuality; empowerment, voice, and choice; and cultural, historical, and gender issues [10]. These principles ensure that the human services are trustworthy and person-centered, targeting trauma survivors’ needs.

The implementation of TIC involves a vital cultural shift, where changes must occur across settings, disciplines, and service users, at all levels of an organization [16,17,18,19]. This includes changing the dynamics of human services so that they (1) realize the widespread impact of trauma and understand potential paths for recovery; (2) recognize the signs and symptoms of trauma in clients, families, staff, and others within the system; (3) respond by fully integrating knowledge about trauma into policies, procedures, and practices; and (4) actively resist re-traumatization (commonly referred to as the four “Rs”) [10]. When these principles are implemented, organizations will shift from viewing social or behavioral issues as problems (“What is wrong with you?”) to seeing them as signs of potential trauma requiring supportive intervention (“What happened to you?”). As part of this TIC approach, relationships will be shaped within a therapeutic community that makes it possible to encourage and coach patients to express their feelings [20].

TIC is distinct from trauma-specific treatment (TST) (such as prolonged exposure therapy or trauma-focused cognitive therapy), which is designed to target trauma-related symptoms or PTSD. TIC is an organizational framework that addresses human service organizations’ cultures and practices in the entire organization and across management levels, where the different levels influence one another and vice versa. As such, TIC implies vigilance in preventing and avoiding institutional processes and individual practices that risk re-traumatizing individuals who have previously experienced trauma.

Implementing TIC is not only important for personal wellbeing and adjustment; it may also play a central role in reducing violence and coercion in mental healthcare settings. This is evident in the six core strategy studies [21,22,23], which is a public health intervention approach that combines six strategies: leadership for organizational changes; using data to inform practice; workforce development where TIC is central; seclusion and restraint reduction tools; peer roles in hospital settings; and the extended use of debriefing techniques, to achieve organizational learning and process traumatic experiences.

Although TIC is still in its early stages, we identified 23 previous systematic reviews exploring its usefulness in various contexts, including programs for children and youth in schools and juvenile justice systems [24,25,26,27,28,29,30,31,32], mental healthcare settings [19,33,34,35,36], and somatic healthcare settings. Two of these reviews were primarily concerned with instruments or measures used to evaluate TIC [37,38], and two focused on TIC training programs for care providers/staff [39,40]. We identified one review focusing on the implementation of TIC at organizational levels across healthcare settings [41], as well as one targeting trauma-specific approaches to treatment across general healthcare settings [42]. The latest review was finalized in 2023 [35] and identified types of trauma-informed approaches in mental healthcare. Several of the previous reviews highlight the potential of TIC for improving mental health and wellbeing outcomes [30], pointing to factors such as senior leadership commitment, sufficient staff support, and the alignment of policy and programming with trauma-informed principles as important for successful implementation [29,41]. However, they also point to weaknesses in the research, including a lack of consensus on the definition of TIC [37], inconsistent methodologies for examining TIC, and various differences in implementation procedures. For example, in some organizations, frontline workers provided grassroots training, while in others it was driven by management [41].

None of the 23 reviews provided a broad overview of the studies carried out across healthcare settings and other human services. Most previous reviews focused on TIC within a specific setting, such as with child- and youth-serving sectors [32], or had a narrower focus, such as on staff training [40] or acute psychiatry [19]. Only one review was conducted in collaboration with trauma survivors [35], and one reviewed nationwide efforts for trauma-informed care implementation [41] and was mainly interested in the first implementation steps and did not capture the long-term effect. TIC has been put forward as a promising approach to solving problems in human service settings, such as psychiatry and the use of coercive measures and violence; however, the existing reviews do not apply a broad overview of how TIC has been utilized across settings and populations. This is needed to establish the current state of knowledge and to guide future decisions about implementing TIC in organizations and countries.

### Aims and Objectives

Our objective was to conduct a scoping review that explores and systematically maps the research conducted in this area and identifies existing knowledge gaps. Our overall aim was “to scope the literature for what is known about implementing TIC?”. More specifically, we sought to answer this question by (1) describing study characteristics, including population, setting, and methodology; (2) synthesizing TIC interventions applied and measures used to examine TIC outcomes as they are applied in those who utilize services and organizations; and (3) identifying key concepts and definitions within the research. We included studies that have implemented TIC across human service settings. We were also specifically interested in mental healthcare, as we expected trauma awareness to be particularly relevant within this field, which is reflected in our search strategy (see below).

## 2. Methods

This scoping review was registered within the Open Science Framework (registration number https://doi.org/10.17605/OSF.IO/RZSKQ, last updated 20 December 2023). A scoping review methodology was chosen [43] as this type of review is appropriate for mapping diverse sources of evidence that underpin complex interventions, such as TIC. The PRISMA-ScR [44] reporting guideline procedure was followed and included 5 steps: (1) formulating the research question(s); (2) identifying relevant studies; (3) selecting studies; (4) mapping data; and (5) collating, summarizing, and reporting the results.

To guide future decisions about implementing TIC in organizations and countries, we found it important to engage with knowledge users, which is also recommended by Pollock et al. [45]. As TIC influences cultures and practices within organizations, we engaged with a Danish National TIC Network that includes stakeholders from different mental health organizations across different management levels and with individual needs and wishes. We assumed they would contribute relevant questions to be considered in the literature review. To avoid tokenism in our engagement of stakeholders, we brought them in early in our study [45], before Step 1, to ensure that the aims of the study reflected their needs. The stakeholders (N = 16) were service users (trauma survivors), staff, managers, representatives of NGOs (non-governmental organizations), and researchers. The stakeholders had different motives for participating in the research. Experts by experience (trauma survivors) were motivated by the ability of TIC to give a voice to survivors and change the culture in human service systems. Experts by profession (mental health practitioners) wanted to learn more about the TIC approach, and managers wanted to know the effects and benefits of TIC.

The majority were clinicians in mental healthcare settings who were planning to become trauma informed. They had different experiences related to trauma, organizations, and mental health, and they were invited to raise questions that were important to them based on their practical or personal experience. In this way, the research questions, as well as decisions about which data should be extracted, were formulated and chosen based on the combined practical and research-based knowledge of the stakeholders and the authors.

### 2.1. Formulating the Research Question

The research questions were formulated with respect to Population, Concepts (such as how TIC was defined), and Context (setting), which are recommended for a scoping review [43]. However, stakeholders influenced the process by, amongst others, raising questions related to paradigmatic changes and influence on empowerment and wellbeing. Together, we decided to focus on those who utilize services, service users, next of kin, and staff (Population); methodologies, TIC interventions applied, and measures used to examine TIC outcomes; and Context, such as organizations and settings. Furthermore, given the lack of consensus and uncertainty surrounding the definition of TIC, we also aimed to map key concepts and definitions within the research, such as how the principles of TIC were incorporated.

### 2.2. Identification of Relevant Studies

A systematic literature search was completed in June 2019, with a follow-up search in February 2022. The search was conducted on seven databases (CINAHL, Cochrane, Embase, ERIC, Medline, PsycINFO, and Web of Science). We also searched gray literature (BASE Bielefeld Search Engine, Google Scholar, and WorldCat); however, due to the vast number of references, we focused on peer-reviewed studies. The search strategy consisted of the following search terms: trauma-informed or six-core strategy, and psychiatry/mental health.

An example can be seen in Table 1.

### 2.3. Eligibility Criteria

The inclusion criteria for the studies were initially developed using a randomly selected subset of studies; however, they were modified by the authors several times throughout the process, in accordance with a scoping review. For example, we excluded gray literature when we identified the huge number of peer-reviewed papers. The inclusion criteria were (a) studies that reported on the implementation of trauma-informed care or the six core strategies; (b) studies reporting any quality improvement related to TIC; and (c) those published in English or a Scandinavian language. Searches were limited to peer-reviewed publications from the years 2000–2022. See Table 2 for a full overview of the inclusion and exclusion criteria.

### 2.4. Selecting Studies

The results of the search were exported into Covidence [46], and duplicates were removed. Covidence was chosen as the reference management program to make the process of inclusion and exclusion more transparent and to increase the review’s reliability. To identify eligible studies, the titles and abstracts of all identified studies were screened first, and then the full text of studies meeting the eligibility criteria was screened. All articles (title/abstract and full text) were screened by two of the authors independently, and in the event of any disagreement about inclusion, the article was discussed with a third author until a consensus was reached. In accordance with the principles of a scoping review [43], no quality assessment of the included studies was performed. Following this procedure, a total of 157 empirical studies were included in the review; the study selection process is summarized in a PRISMA flowchart in Figure 1. A total of 23 systematic reviews concerning the implementation of TIC were identified through the process and were checked for missing citations; they are included in Section 1 and Section 4. Two of these reviews were published after the search.

### 2.5. Mapping Data

Data were extracted from the studies by pairs of authors into a Covidence data charging form that was developed based on the initial consultation workshop with stakeholders. The data were mapped into the following categories:Study characteristics (population studied, setting, organizational level);Research methodology/study design (e.g., quantitative, qualitative);Definitions of TIC;TIC interventions used, including training programs;Outcomes and measures used to evaluate the effect(s) of TIC.

Any disagreements between extractors were resolved through discussions among all of the authors. In order to ensure consistency and credibility in the extraction process, we developed a document in which we highlighted the agreed-upon understandings of core concepts and we also held numerous meetings.

### 2.6. Collating, Summarizing, and Reporting the Results

To address the aim and objective based on the full set of 157 studies identified in the scoping review, we summarized the numerical data and carried out a narrative description of the textual data.

## 3. Results

This section is divided into three subsections. First, we provide an overview of the characteristics of the included studies and study settings and their methodologies; second, we summarize the key concepts and definitions used across studies; and finally, we describe the interventions and measures used.

We found an increase in the publication rate of studies meeting the inclusion criteria from 2015 until the termination of the inclusion period in February 2022 (see Figure 2). While the number of studies is likely to increase further, we chose not to repeat the search, as the included studies showed great diversity and the addition of more studies would likely not have affected the purpose of our study, as we had not identified any new themes in the updated search in 2022.

### 3.1. Study Characteristics

This scoping review includes studies with different objectives, conducted in various countries and settings (primary care, specialty care, dentistry, emergency treatment) at different organizational levels. They each have unique characteristics and challenges, depending on the local context of implementation. Thus, a structured presentation of these studies is crucial to facilitate the analysis. We organized the studies based on their country of origin, organizational level, setting, and methodology.

Country: The majority of the studies were conducted in the USA (*n* = 107, 68%), along with other English-speaking countries, including Canada (*n* = 18, 11%), Australia (*n* = 13, 8%), and the UK (*n* = 7, 4%). Seven studies were conducted in Germany (*n* = 3, 2%), Finland (*n* = 1, 0.6%), Greece (*n* = 1, 0.6%), and Japan (*n* = 1, 0.6%). Finally, one study (0.6%) took place across five countries, including the USA, Northern Ireland, Israel, and Trinidad and Tobago [47].

Organizational Level: Approximately 44% (*n* = 70) of the studies were conducted at the regional level, spanning a state or a larger region of a country; 15% (*n* = 23) were conducted at the hospital level, 13% (*n* = 21) at the national (country) level, and 13% (*n* = 21) at the county/city level. About 9% (*n* = 14) were conducted at units within hospitals, and 5% (*n* = 8) were in educational health facilities or social work settings (*n* = 1, 0.6%). 

Setting: Nearly 30% of the studies were conducted within child welfare or educational settings (*n* = 47), including children’s mental health services, behavioral health agencies, medical centers, residential foster care and/or treatment facilities, juvenile justice systems, educational institutions, nonprofit organizations, and private case management agencies, among others. Child welfare services covered those aged from 3 to 25 years in the included studies. Cross-sectoral studies (*n* = 27) made up 17% of the study settings, integrating primary and secondary care or behavioral health and community services, among others. Twenty-one studies (13%) took place in mental health and/or behavioral health settings, with the majority being somatic healthcare settings (*n* = 19, 12%), for example, emergency departments, community healthcare, and pediatrics. TIC was implemented in healthcare or social work education settings, such as nursing and medical schools, in 11 studies (7%). Nine studies (6%) were conducted in residential treatment services, and the same number within social services. Finally, four studies were carried out in substance abuse services, and four were conducted in intellectual and developmental disability services. Seven (4%) of the studies did not fit into any particular setting, including activities such as church reintegration and the fostering of children in Rwanda or women exiting prostitution [48,49].

Methodology: The majority of the included studies applied a quantitative design (*n* = 70), accounting for 44% of the studies. The remainder were divided into qualitative (*n* = 36, 23%) and mixed-method (*n* = 31, 20%) studies. Quality improvement was used in 13% of the studies (*n* = 20), where naturally occurring data (documented instances of reduced coercion) were utilized and observed over extended periods.

### 3.2. Definitions of Trauma-Informed Care

We reviewed the studies based on their definitions of TIC because conceptualizing what is meant by TIC is an important first step to operationalizing and implementing it. A majority (*n* = 146) of the studies included one or more definitions of trauma-informed care as part of their introduction or referred to other sources for definitions (*n* = 6). The most-used definitions were those provided by SAMHSA [10,50], which was used in 73 studies, and Harris and Fallot, for example, [15,51], which was used in 41 studies. Bloom’s sanctuary model was used as a definition in 16 studies, for example, [52]. All three of these models focus on a set of concepts or principles to which organizations should adhere in order to be trauma informed (see Table 3 for an overview of the key concepts). While the different definitions vary, they also have commonalities. For example, safety is a key principle highlighted in all of these models [10,15,52]. Although most of the studies referred to SAMHSA’s definition, including the six core principles (safety; trustworthiness and transparency; peer support; collaboration and mutuality; empowerment, voice, and choice; cultural, historical, and gender issues), the studies varied greatly in how they applied this definition. For example, some of the studies did not apply any of the principles or focused on only one or two of them.

There were also studies that referred to other models of trauma-informed care in their definitions. For example, twelve studies referred to Elliott, Bjelajac, Fallot, Markoff, and Reed’s [53] ten principles of trauma-informed care, four studies mentioned Chadwick’s trauma-informed system project [54], and four studies referred to the work of Hodas [55] on trauma-informed care in child and youth welfare. Sixteen studies used previous reviews of trauma-informed care in their definitions, seven studies referred to a review by Muskett from 2014 [33], and nine studies used Hopper and colleagues’ overview of definitions from 2010 [56]. Finally, there were three studies that based their definition on the six core strategies [23]. It should be noted that definitions that were used in only one or two of the included studies are not mentioned here.

### 3.3. Interventions

The studies included several different and overlapping interventions or processes designed to implement TIC. Interventions were understood here as “…a combination of activities or strategies designed to assess, improve, maintain, promote, or modify health among individuals or an entire population…” [57]. The majority of the studies included some form of intervention, and more than half focused on educational interventions, with only a few studies applying the same teaching materials. Thus, the educational interventions varied greatly, as the length of the education programs ranged from 30 to 60 min of online training [58,59] to training lasting for several years [60]. To obtain an overview, we identified three categories of interventions: (1) educational interventions alone; (2) other interventions; and (3) educational interventions and other interventions. We also included a fourth category, (4) no interventions.

Educational intervention only: These were studies that only described an educational intervention, such as training staff members in learning how to ask [61], or a one-day trauma awareness course [62]. An example is Trauma Smart training [63], which is an organizational change intervention/curriculum designed to build trauma-informed knowledge in schools. It includes 10 modules: (1) Why Become Trauma Smart?; (2) Developing a Common Language; (3) Caregiver Affect Management; (4) Attunement; (5) Routines and Rituals/Consistent Responses; (6) Affect Identification; (7) Affect Modulation/Affect Expression; (8) Grief and Loss; (9) Executive Function; and (10) Self-Development and Identity/Trauma Integration.

Other interventions: These were studies that implemented non-educational interventions, such as trauma-specific interventions, which included trauma screening [64], therapeutic activities, supervision [65], or mindfulness and engagement training [66,67]. “Other interventions” also included organizational and environmental interventions, such as breakthrough methods and PDSA circles [16,68,69,70] and curriculum development [71,72].

Educational interventions and other interventions: These were studies that combined the two previously mentioned categories. The combinations of interventions were informed by specific TIC approaches or programs and definitions of TIC (see Table 3). For example, if the implementation was guided by the SAMHSA definition [10], it would consist of trauma screening, education, supervision, and other organizational changes. An example of combined intervention is the work of Barton et al. [71], who used the six core strategies, environmental changes, curriculum development, and staff training [71].

No intervention: There was also a relatively high number of studies (*n* = 45, 28%) that did not include any intervention. These were studies about how certain measurements were validated, such as [70], which aimed to validate TICOMETER [70], a five-domain measure of the organizational implementation of trauma-informed care that can be used to evaluate cultural changes. Survey studies, or qualitative studies—for example, those examining users’ perceptions of or experiences with the implementation of TIC—were also included here [73], as were quantitative studies that reanalyzed existing data from previous studies [74].

### 3.4. Outcomes and Measures Used to Evaluate the Effects of TIC

Several outcomes and measures were used to evaluate the effects of TIC, and we categorized them based on the individuals, groups, or organizations to which they were applied, resulting in three target groups: (1) service users, which included trauma survivors or patients who were recipients of service; (2) service providers, who were staff members and a part of the service delivery; and (3) organizations. Because it was sometimes difficult to distinguish between outcomes and measures for service providers and organizations, we decided that those involving service providers’ evaluations of their own experiences or knowledge, levels of stress, or attitudes and knowledge about TIC, were grouped under the heading service providers. Measures related to service providers’ or service users’ evaluations of organizational issues (resources, support, or evaluations of the success of the implementation of TIC) were grouped under organizations. Some could be placed in both categories, such as work environment scales. In these few cases, after a consensus discussion, the author group decided which group was the most appropriate. We placed more emphasis on trauma and PTSD measures in the review, as we saw them as especially important when examining TIC.

Recipients of service: In studies that included clinical measures to examine recipients of service, trauma and PTSD screening tools were the ones most frequently applied. The majority of these studies used validated trauma measures, non-validated scales, administrative data, or a combination of sources. However, some studies did not report which measures were used or employed self-generated questions.

Measures that were used in child and/or adolescent populations included the Young Child PTSD Checklist “YCPC” [75], the University of California, Los Angeles “UCLA” and Posttraumatic Stress Disorder Reaction Index “UCLA PTSD-RI” [76], the Child Trauma Screen “CTS” [77,78], formerly called the Connecticut Trauma Screen, the Trauma Symptom Checklist for Children “TSCC”, The Child PTSD Symptom Scale [79], the TIC Grade [80], and the Screen for Child Anxiety-Related Emotional Disorders “SCARED” [81,82]. There were four studies that identified trauma exposure in children/adolescents based on administrative data and maltreatment reports (physical abuse, neglect, sexual abuse) extracted from child welfare administrative databases [83,84,85,86].

Measures that were used to examine trauma exposure and/or reactions in adult populations included the Adverse Childhood Experiences (ACEs) Questionnaire [87], the Posttraumatic Diagnostic Scale [79], the PTSD Checklist “PCL” [88], the Global Assessment of Individual Needs “GAINS” [89], the Psychological Maltreatment of Women Inventory “PMWI” [90], the Conflict Tactics Scale [91], the Sexual Experiences Survey “SES” [92], and the Identity Abuse Scale [93]. To the best of our knowledge, all of the measures mentioned in the previous two sections, except for the “YCPC” [75], have been validated. However, a short version of the “YCPC” has subsequently been validated [94].

While trauma and PTSD screening tools were the ones most frequently applied among service users, there were a range of other measures used to evaluate the implementation of TIC, and several studies included more than one measure. We divided them into two categories. The first category included data that were observational in nature and primarily collected as part of routine documentation or registration in clinical practice. The second category included measures used to examine service users’ health and wellbeing, either as evaluated by service providers or as reported by the service users themselves. These two broader categories were then further divided into smaller categories or themes based on the types of outcomes that were measured (coercion/aggression management, mental health, or physical health outcomes). We only formulated themes if we could include at least three studies. The themes were formulated via a discussion between two of the authors (J.H. and T.H.). The themes in the first category were “Aggressive behavior” and “Coercion/aggression management”, while those in the second category were “Mental health symptoms”, “Physical health”, and “Self and Well-being” (see Table 4 and Table 5).

Service providers: Most of the studies used questionnaires to assess staff knowledge and attitudes in relation to TIC training (see Table 6). Many [63,95,115,116,117,118,119,120,121,122,123,124,125,126,127,128,129] used the Attitudes Related to Trauma-Informed Care (ARTIC) scale, which has been validated in different versions examining the attitudes, knowledge, practice, and competencies of service providers [130,131]. ARTIC is a self-report instrument for staff across human service systems and educational settings, and it can inform individuals and organizations about attitudes that are less favorable to TIC and, as a result, provide suggestions as to where and how training should be targeted.

Finally, seven studies examined secondary traumatic stress in service providers (child welfare workers, healthcare providers) using the Professional Quality of Life Scale [177].

Organizations: Several studies examined barriers and readiness to implement TIC, the relevance of TIC programs, and implementation success at an organizational level. We found four validated instruments assessing organizational trauma-informed care status and changes (see Table 7).

An example is the Trauma-Informed Self-Care Revised (TISC-R), which consists of three latent factors: (1) utilizing organization resources and support; (2) organizational practices; and (3) personal self-care practices. The brief TISC-R measures trauma-informed self-care practices and can help assess high-stress environments [145]. There were also studies evaluating trauma-informed practice in organizations based on the work of Harris and Fallot (2001) [15], where assessments focused on five core values of safety, trust, choice, empowerment, and collaboration; for example, issues related to empowerment revealed how the strengths and skills of staff are utilized and developed in organizations, such as if leadership recognized the strengths and skills of employees [106,112,178,179,180].

A range of studies used qualitative methods to assess the influence of TIC on service providers’ cultural experiences [17,111,148,150,151,152,157,160,161,166,173,175,197,198,199,200,201]. Additionally, organizational factors were assessed using unvalidated surveys in several studies in which the surveys were invented for those specific studies.

Finally, service users evaluated the degree of implementation of TIC or described what challenges organizations face when trying to implement TIC, specifically as seen from the perspective of service users (see Table 8).

For example, in Kusmaul et al.’s study [143], service users found it challenging for agencies to provide trauma-informed care (TIC) concepts for all at the same time.

## 4. Discussion

The overall aim of this review was to gain knowledge about how trauma-informed care (TIC) has been implemented across human services. We provided an overview of 157 studies identified from more than 3000 papers, focusing on their settings, methodologies, and definitions of TIC, as well as the types of interventions and measures used. Such an overview is important to establish the current state of knowledge about TIC and to facilitate decisions about how best to implement and adopt the approach. Even though several previous reviews exist, our review provides a more comprehensive overview of the literature, as we did not limit our search to specific settings or populations. Furthermore, this review was developed in close collaboration with stakeholders, ensuring the relevance of the research to policymakers, researchers, and others who have the power and responsibility to change mental healthcare and other service systems.

Stakeholders who were experts by profession, experts by experience (trauma survivors), managers, or researchers helped to identify the research questions and the data to be extracted. Thus, multiple perspectives and types of knowledge shaped this review. There is limited knowledge about how this influences the research process; however, user collaboration in research is important in achieving societal impact, and this marks a shift in approach from assuming that the researcher is the only expert to a new paradigm where researchers and knowledge users are both viewed as experts bringing complementary knowledge and skills to the team [202], in line with the principles of TIC. The stakeholders had different motives for participating in the research, and different stakeholders wanted answers to different questions—for example, how outcomes such as wellbeing and coping were related to TIC, or which paradigmatic changes were documented in the literature. Some of these initial and personalized objectives were difficult to accommodate, for example, because variables were not included in studies or were only mentioned implicitly. Furthermore, most of the stakeholders did not have extensive knowledge about TIC at the beginning of the study, limiting the types of questions that could be formulated. Therefore, the data extraction was both deductive and inductive, where data were searched for based on predefined questions while also leaving room for information accumulated from the data, such as whether or how we should extract data from educational interventions at higher educational institutions [141], and then we began to look for students as a category. Muddling through these complex demands and expectations, according to Mead [203], can foster tension between members of a research group, leading to confusion about which data to extract. However, recalling the purpose of a scoping review—as a mapping exercise and not as a synthesis of the best available evidence—helped us to focus this work. By rigorously defining the inclusion and exclusion criteria, as well as which data should be extracted (see Appendix A), it was possible to extract relevant information from the studies. In the following sections, we discuss findings related to the study characteristics, as well as the definitions, interventions, and outcomes used.

The majority of the studies were from the United States, where attempts have been made for several years to increase awareness of trauma in vulnerable populations—such as children in welfare programs, individuals with mental and/or substance use disorders, and prison inmates—and implement trauma-informed care at all organizational levels. Similarly, in Australia, the Mental Health and Suicidal Prevention Plan states that staff must be trained in TIC, and the NSW Department of Health has several policies that mandate services to reflect the principles and values of TIC [36]. However, there is a paucity of studies implementing TIC in other countries, e.g., in Europe, although there is growing interest in this area due to its high success rates in reducing coercive practices [65]. This is very much needed according to the European Parliamentary Assembly [204], which demands that psychiatry ends coercion and calls for a human-rights-based approach, which TIC is.

Most of the studies were conducted in child/adolescent services or educational settings. This seems meaningful, as early preventive efforts can not only avoid human suffering but also reduce societal costs. For example, the Institute for Trauma and Trauma-Informed Care [205] has estimated that child abuse costs more than USD 5 trillion, and the Washington Family Policy Council states that it is possible to save USD 1.4 billion over a decade if TIC is used in social services and schools [206]. However, no studies have focused specifically on the cost-effectiveness of TIC, except for Saunders et al. [35], who mentioned how the implementation of TIC is long-lasting and expensive.

While around 30% of the studies focused on services and education for children, they were conducted in diverse populations and settings, using a myriad of methods. All of them adapted the TIC approach in unique ways to fit specific contexts. On one hand, this speaks to the advantage of TIC as a broad and holistic approach to services. Because trauma can affect everyone, the approach needs to be applicable across a wide range of settings. However, on the other hand, it can be a disadvantage if the approach is implemented in varied ways because this makes it hard to compare findings and thereby determine the effects of TIC (it could be that the elements are helpful, but for whom, and why?). Furthermore, and more fundamentally, it is difficult to determine whether studies are in fact examining the same phenomenon if they define and operationalize TIC in different ways.

While most of the studies mentioned at least one of the six principles outlined by SAMHSA [10] (safety; trustworthiness and transparency; peer support; collaboration and mutuality; empowerment, voice, and choice; and cultural, historical, and gender issues) when defining TIC, there was a high degree of variation in how these principles were applied. For example, while some studies implemented all six principles, others focused on one or two, potentially because it was difficult to implement all principles at the same time or because some principles were viewed as more fundamental to TIC than others. This could be why safety was mentioned more frequently than the other principles. Another challenge when implementing the principles of TIC is that they are conceptually ambiguous and open to interpretation. For example, some studies focused on physical safety but not psychological safety, or vice versa. Another example is the principle of empowerment, which can be viewed as a process or an outcome and as an internal experience or as something occurring between individuals, making it difficult to measure. While the implementation and evaluation of the principles of TIC were described as fundamental to becoming trauma informed in some studies [133], other studies focused on the effects of training programs that teach healthcare professionals about the concept of trauma and its consequences [64,157] or on the effects of integrated trauma-informed services that include trauma-specific screening and treatment, for example, [49,83], thus paying more attention to the “4 Rs” (realize, recognize, respond, resist re-traumatization) in their operationalization of TIC. However, these studies were equally heterogeneous; for example, in the contents of the included teaching materials. Finally, there were some who created their own working definitions of TIC, such as [207], p. 10, who conceptualized TIC as follows:

“Care which occurs in a trauma-informed environment where universal precaution exist, in which the trauma-aware nurse, displaying empathy and strength based action, delivers care resulting in empowerment, relationship—building, and resilience in the form of reduction in trauma triggering”.

This conceptualization does not capture a multidisciplinary approach, as it is restricted to nursing and is grounded in a value-based approach.

Thus, our review shows that there is a need for a narrower and less ambiguous definition of TIC and a higher degree of international consensus about how TIC should be operationalized. More consistent terminology would enhance transparency and simplify communication among end users, including policymakers and researchers.

It was not just how definitions were applied that varied across studies but also which interventions were used to implement TIC. Most were based on educational interventions, with studies using different curricula, making it difficult to infer which were more effective. This is similar to a review by Gundacker et al. [39] that mapped 17 TIC curricula for outpatient settings in different categories, including person-centered communication skills, understanding the health effects of trauma and collaboration, understanding one’s professional role, understanding one’s own history, and screening. While they revealed several positive reactions, including improved knowledge, attitudes, trauma screening, and communication, they did not identify changes in health outcomes for service users, staff satisfaction, or organizational improvements. Some curricula were tailor-made to fit specific contexts or goals, e.g., PARS [18,104], which included interventions designed to build capacity to reduce coercive measures, such as de-escalation and debriefing training. This is a fundamental part of TIC, which gives control back to patients and avoids re-traumatization. According to Sweeney et al. [11], interventions that mitigate potential sources of coercion and accompanying triggers may be preferred, and the organization must also be aware of how helping can enforce helplessness and shame. Tailor-made curricula reflect the specific needs of target groups; however, this makes it difficult to synthesize knowledge across studies.

TIC involves a vital cultural shift for organizations. Therefore, moving TIC into practice demands changes at many different levels in the organization and the hierarchical structure [18]. This was one of the reasons why we involved stakeholders in this review, as they guided the questions in this direction, and this led to the decision of the broad scope across settings considered in our study. This broad scope might be helpful during long-term implementation [45]. However, this cultural shift can be challenging, as emphasized by Wilson et al. [36], who highlighted how internal structures can counteract the purpose of TIC, such as when the treatment is fragmented and not facilitated through cross-agency coordination [15]. Even though our main focus was on mental healthcare, the inclusion of TIC across settings raised an awareness of why TIC cannot solely be adapted in one setting but instead must go across agencies. According to Bloom [208], organizations are living organisms, where the management system influences the behavior and the functional system (staff members), and vice versa. The organization will constantly interact with and be created by interactions with others, and it is shaped by leadership behavior. This will influence the service, which within a trauma-informed approach will promote growth and mastery [15], while within a traditional cost-conscious environment where treatment is defined narrowly, it will promote an atomistic approach, where life stories are neglected. Within this traditional approach, there can be a reluctance to shift from the biomedical causal models to a holistic biopsychosocial model of mental distress, which is a barrier to becoming trauma informed [11].

Because TIC involves changing cultures and complex organizational practices, a multifaceted implementation approach could be preferable. However, our review also shows a need for more simplistic and unanimous approaches to implementing TIC, potentially where steps or phases of implementation are more clearly delineated. In addition, TIC initiatives should be informed by implementation science, where the organizational context is taken into account [209]. The different contexts are well described in the studies (as can be seen in Appendix A), but only a few studies have used a specific implementation science framework, such as RE-AIM [122,186].

Finally, several studies did not examine the outcomes of implementing TIC. This is problematic because if the implementation of TIC continues to spread, without any control of the outcomes, there is a risk of interventions having no effect or, worse, producing harm [210]. In studies that evaluated the outcomes of the approach, we sought to meet the triple purpose of TIC in our mapping, including improved health outcomes and wellbeing in service users, improved satisfaction in service providers, and improved organizational structure and delivery. However, there was little consensus on which measures to use to examine outcomes in service users. For example, several different scales were used to identify trauma exposure/reactions and, while most were validated, other measures were non-validated and/or self-generated. This goes against the recommendations of Elliott et al. [53], who argued for the use of validated scales to acquire systematic evidence across studies. Most measures used with service users were related to symptoms, such as PTSD or coercion, thereby potentially overlooking elements related to wellbeing, quality of life, empowerment, and other positive states that are as equally important to recovery as the absence of illness/symptoms. Furthermore, we noted that some studies examining service users’ experiences did so solely based on observations from care providers [148], which, according to Isobel [211], does not align with the expectations of TIC research to be trauma sensitive. This also seems counterintuitive when considering that a basic principle of TIC is giving a voice to survivors. Most measures used in relation to service providers examined their knowledge or attitudes, for example, before and after TIC training, or their evaluations of TIC’s implementation, potentially overlooking their trauma and trauma reactions. It is important to ensure a higher degree of agreement and consistency in terms of which measures are used to examine the outcomes of TIC. However, it is also relevant to consider that research on TIC must be conducted in a trauma-specific manner [211], including trauma survivors’ participation in the research and adherence to certain ethical guidelines, such as ongoing consent checking. Additionally, the research must go beyond whether it works in the sense of achieving an intended outcome (for example, lowering the use of coercive measures) and ask broader questions, such as how it contributes to systemic change or how evidence can be used to support real-world decisions. The Medical Research Council updated its framework for developing and evaluating complex interventions [212] with advice on how a system perspective can be helpful in implementing complex interventions, such as TIC. Our scoping review has identified a myriad of possibilities and has answered questions in collaboration with those who utilize these services, which could represent a first step in identifying interventions appropriate for the sustainable implementation of TIC.

### Strengths and Limitations/Reflexivity

Several limitations of this review must be considered. This was an enormous undertaking, and our results are only up to date until February 2022; several other reviews have been conducted both before and after this date, such as the one by Saunders et al. [35]. A more systematic approach to these reviews could have delivered a comprehensive overview of the available evidence, but this would not have answered all of our research questions, nor would it have captured the vast variation in how TIC is defined, operationalized, and measured across studies. This complexity is not captured by looking at previous reviews because they focus on selected studies or isolated aspects of TIC, for example, staff training programs. Many researchers were involved in our review process and combined with the blurry and varied descriptions of TIC’s key concepts and definitions, this made it difficult to determine a clear direction. The analysis required the authors to take additional steps to synthesize and draw useful conclusions from the studies considered. The reporting of the studies was very diverse and carried out in an unsystematic manner; it was also necessary to retrieve the papers several times to ensure that all data were captured. The multidisciplinarity in the research group (the authors of this paper), along with the thorough screening and mapping process, involving all members in decisions until consensus, demonstrates rigor and quality, as well as reflexivity [213], where we as researchers acknowledged our role in the research. This was an awareness of how our prior experiences as psychologists and nurses and our assumptions and beliefs could influence the research process.

Aside from the work of Isobel [211], there exists limited studies to indicate the trustworthiness of the findings. The author details the importance of reflexivity, such as an awareness of what the researchers bring to the study and which norms guide the analysis of the research process. The participatory approach sought to establish fairness through the criteria described by Lincoln and Guba [214], where knowledge users’ inputs are incorporated into the process, and they were involved in the formulation of the research questions and the mapping of the data. This democratic process is consistent with the core idea of TIC—“nothing about us, without us”—but the long duration of the process caused us to occasionally lose sight of stakeholders’ fundamental ideas. However, initiating the research with a group of knowledge users who had experienced trauma in their lives ensured this focus in our review. Practical knowledge from everyday life was considered, but it also complicated matters, as several new questions were raised, such as how paradigmatic changes and mastering have been addressed in existing research.

## 5. Conclusions

The experience of trauma is widespread across the world; it is linked to a range of poor social and health outcomes that incur substantial costs to individuals and societies. The six guiding principles of TIC ensure that individuals are met, treated, and appreciated in the context in which they are living their lives. However, our review shows that this is a complex and multifaceted framework, with no overarching structure or clear theoretical underpinning that could guide research and practical implementations. It has been defined and adapted in varied ways across different settings and populations, using different interventions and outcomes, making it difficult to synthesize the knowledge. However, there is an overall agreement that TIC is a whole-system approach that illuminates and tries to counteract traumatizing practices, where the risk to trauma survivors is weighed alongside risks to providers. TIC aims to identify capacities in people and to prevent adverse outcomes by addressing root causes, such as adverse childhood experiences, for example, by applying trauma screening practices. Therefore, TIC seems to be a necessary approach to healthcare and social services, along with public health. TIC is still a relatively new concept, and it might still be in a pre-paradigmatic stage, in which a common language is developed; hence, more comprehensive research related to TIC is needed. In the future, certain conditions must be in place to guide decisions about how to implement TIC in organizations. Foremost, this must be carried out together with people who have the power and responsibility to decide how to implement and how to measure the effects of TIC. This can provide a roadmap with which to radically rethink psychiatry and build a service system with the participation of all stakeholders, including in particular persons with mental health conditions and service providers. TIC is gaining momentum, and several new studies and reviews [35,207] have been published since this scoping review was started. This is because it resonates with human beings’ expectations of what service systems should be like and because it resonates with a human rights-based approach [215]. We advocate for a trauma-informed approach to healthcare as a standard of care due to the sensitive nature of mental health issues, as well as the likelihood that many patients may not disclose their trauma history.

## Figures and Tables

**Figure 1 healthcare-12-00908-f001:**
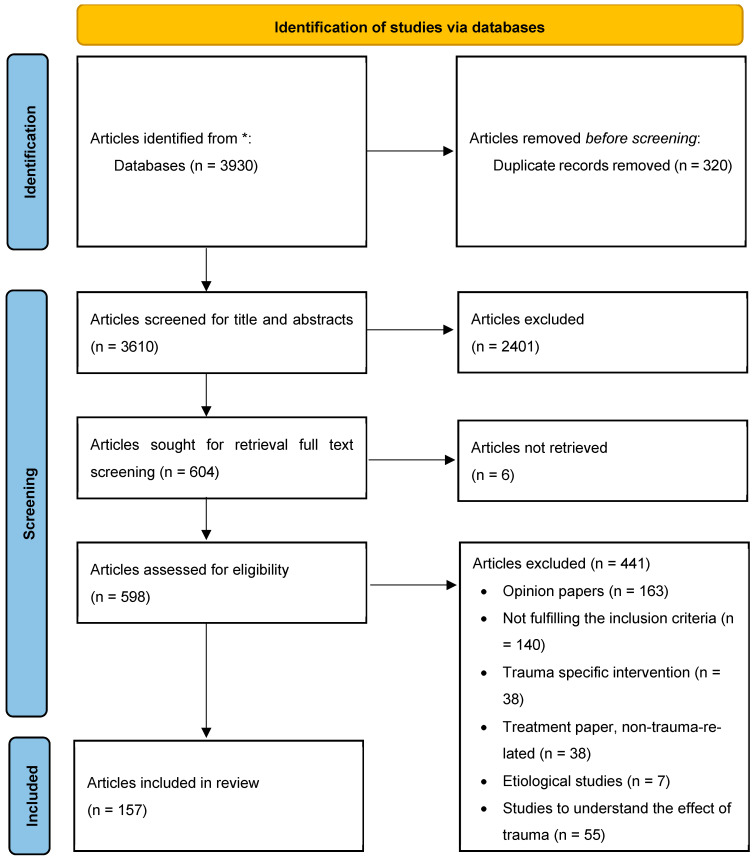
The PRISMA flowchart of the literature search. * CINAHL, Cochrane, Embase, ERIC, Medline, PsycINFO, and Web of Science.

**Figure 2 healthcare-12-00908-f002:**
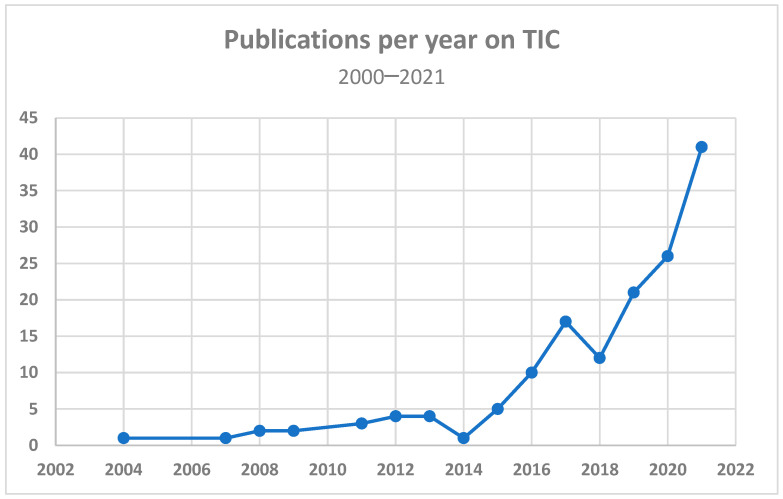
Publications per year among the included studies.

**Table 1 healthcare-12-00908-t001:** Example of the search strategy, * all truncation used.

CINAHL via EBSCO, Wednesday, 23 February 2022, 12:23:13 p.m.		
S1	TI trauma-informed OR AB trauma-informed	1.711
S2	TI six-core strateg* OR AB six-core strateg*	17
S3	(MH “Mental Health”) OR (MH “Mental Health Services+”) OR (MH “Mental Disorders+”) OR (MH “Psychiatry+”) OR (MH “Psychiatric Units”) OR (MH “Psychiatric Service”) OR (MH “Psychiatric Patients+”)	714.025
S4	S1 OR S2	1.725
S5	S3 AND S4	812
S6	S3 AND S4	302

**Table 2 healthcare-12-00908-t002:** Inclusion and exclusion criteria.

Inclusion	Exclusion
Published in the year 2000 or later Includes the words, “trauma-informed” OR “trauma-sensitive” OR “trauma approach” OR “Six core strategies” Can be a description of use within practice in any way Any country In English, Danish, Swedish, or Norwegian Peer-reviewed	Theses with articles published Conference paper linked to a full article Describing trauma-specific long-term interventions Systematic or other types of review Etiological studies Treatment papers (specific, not-trauma-related, interventions) Study protocols Purely theoretical paper If TIC is only mentioned in perspectives

**Table 3 healthcare-12-00908-t003:** Overview of key concepts used in different definitions of trauma-informed care (TIC).

Key Concepts	Definitions			Number of Studies Mentioning Key Principles N (%)
	Substance Abuse and Mental Health Services Administration	Harris and Fallot	Bloom’s Sanctuary Model	
Safety	X	X	X	113 (71%)
Trustworthiness and transparency	X	X		94 (59%)
Peer support	X			69 (43%)
Collaboration and mutuality	X	X		105 (66%)
Empowerment, voice, and choice	X	X		107 (67%)
Cultural, historical, and gender issues	X			93 (59%)
Sensitivity to trauma		X		-
Non-violence			X	-
Emotional intelligence			X	-
Democracy			X	-
Open communication			X	-
Social responsibility			X	-
Commitment to social learning			X	-
Growth and change			X	-

Note: For the “Number of studies using key concepts” column, we focused only on the 6 key concepts defined by SAMHSA, as this framework was used most frequently. However, studies mentioning key concepts (e.g., peer support) without referring to the SAMHSA definition were also included in the count.

**Table 4 healthcare-12-00908-t004:** Overview of observational data collected as part of routine documentation or registration.

Examples of Outcomes	Examples of Measures	Examples of Studies
Aggressive behavior, aggression, challenging behavior, staff injuries	Agency-developed form [1,95] Hospitals’ registers/databases [96] The Child Behavior Checklist [97]	[1,95,96,97]
Coercion/aggression management, restraint, seclusion, room observation, timeouts, PRN medication	Agency-developed form [1] Audits [98] Electronic medical records [99] Hospitals’ registers/databases [22,65,96,100,101,102] Statewide registers [103,104]	[1,22,65,71,96,97,98,99,100,101,102,103,104,105]

**Table 5 healthcare-12-00908-t005:** Overview of measures to examine service users’ internal experiences.

Examples of Outcomes	Examples of Measures	Examples of Studies
Mental health symptoms, e.g., depression, shame, externalizing/internalizing, drug abstinence rates, emotional health, improvements	The 10-item Shame Subscale of the Personal Feelings Questionnaire–2 (PFQ2-Shame) [106] The Addiction Severity Index [107] The Brief Symptom Inventory (BSI) [107] The Center for Epidemiological Studies Depression Scale (CES-D) [108] The Child Behavior Checklist [97] The nine-item Patient Health Questionnaire [106] The six-item Cognitive Reappraisal Scale of the Emotion Regulation Questionnaire [106]	[97,106,107,108,109,110]
Physical health, e.g., physical health, physiological stress levels	The seven-item Somatization Subscale of the Brief Symptom Inventory (BSI) [106]	[106,111]
Self and wellbeing (broadly defined), e.g., self-awareness, self-determination, self-efficacy, self-worth, wellbeing, coping	Items on the Interpersonal Development Social Outcomes Scale [112] Measure of the Victim Empowerment Related to Safety Scale [110] The Restorative Parenting Recovery Index (RPRI) [113] The six-item short form of the 11-item De Jong Gierveld Loneliness Scale [106] The Subjective Well-Being (SWB) scale [114]	[106,110,111,112,113,114]

**Table 6 healthcare-12-00908-t006:** Summary of staff-related outcomes.

Examples of Outcomes	Examples of Measures	Examples of Studies
TIC knowledge	ARTIC, TIMCQ, TICOMETER, TICQ, TKQ, The Trauma-Informed Belief Measure, Trauma-Informed Knowledge Scale	[1,17,20,22,47,48,49,58,64,66,68,69,71,83,85,86,97,98,100,101,106,107,110,124,130,132,133,134,135,136,137,138,139,140,141,142,143,144,145,146,147]
TIC practice	ARTIC, TIMCQ, TICOMETER, TICQ, TKQ, The Staff Behavior in the Milieu Survey	[59,84,114,124,127,134,135,139,148,149,150,151,152,153,154,155,156,157,158,159,160,161,162]
Staff competencies	ARTIC, TIMCQ, TICOMETER, TICQ, TKQ, TICPS	[59,64,100,140,158,163,164,165,166,167,168,169,170,171]
Staff attitudes	ARTIC, TIMCQ, TICOMETER, TICQ, TKQ, TIP, Trauma-Informed Knowledge, Attitudes, and Beliefs of Providers Scale	[47,58,63,64,74,100,106,111,115,116,117,119,120,121,122,123,125,126,127,128,129,138,149,159,163,168,169,171,172,173,174,175,176]

ARTIC: Attitudes Related to Trauma-Informed Care; TIMCQ: Trauma-Informed Medical Care Questionnaire; TICOMETER: Organizational Trauma-Informed Care in Human Services; TICQ: Trauma-Informed Care Questionnaire; TKQ: Trauma Knowledge Questionnaire; TICPS: Trauma-Informed Care Provider Survey; TIP: Trauma-Informed Practice Scale.

**Table 7 healthcare-12-00908-t007:** Summary of organizational problems assessed by staff or managers.

Outcomes	Examples of Measures	Examples of Studies
Principles of TIC	TISC-R, TISCI	[1,16,18,48,49,60,62,68,70,109,112,118,120,141,178,179,180,181,182,183,184,185,186,187,188,189,190]
Organizational readiness to implement TIC	TSRT	[60,67,84,124,127,138,143,178,190,191]
Was the TIC program relevant to the organization?		[62,66,69,126,127,147,166,179,181,185,187,191,192,193]
Barriers against TIC	TICPS	[60,64,127,142,181,193,194]
Implementation success		[60,132,152,193,195,196]
Trauma-informed practice, e.g., safety, trust, empowerment, choice, collaboration	The Trauma-Informed Organizational Environment Scale Trauma-informed organizational culture measure	[106,112,178,179,180]

TISC-R: Trauma-Informed Self-Care Measures Revised; TISCI: Trauma-Informed System Change Instrument; TSRT: Trauma System Readiness Tool; TICPS: Opinion subscale of the Trauma-Informed Care Provider Survey.

**Table 8 healthcare-12-00908-t008:** Overview of outcomes collected from service users to evaluate the degree to which TIC has been implemented or taken as a focus area for organizations.

Examples of Outcomes	Assessments of Outcomes (Examples)	Example of Studies
The six principles of TIC	Qualitative	[143,150,189]
Implementation level (was TIC implemented as intended?)	Qualitative	[49]
Culture in the institution		[105]

## Data Availability

Data sharing is not applicable to this article, as no new data were created or analyzed in this study.

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
