# Peer review of "Implementing Trauma-Informed Care—Settings, Definitions, Interventions, Measures, and Implementation across Settings: A Scoping Review"

_healthcare, 2024, doi:10.3390/healthcare12090908_

Round 1
Reviewer 1 Report
Comments and Suggestions for Authors
Thank you for this opportunity to review this manuscript. It covers an important topic and a scoping review would improve our knowledged on TIC. The intro is very well written, the PRISMA-ScR is covered adequately. Initially my edits were minor. However, when I got to the Interventions, the writing style, citation style, and font style was very inconsistent. I find this concerning for plagairism or AI. I stopped my review at this point based on my concerns. I made the following notations up to this point:
Line 55 – Would make two sentences. New sentence starting with, “For example, if they…”
Line 70 – I did not see quotes to suppor the “(p. 5)”
Line 145-149 – The text is different in these lines which is concerning for plagiarism. Quotations must be used for direct quotes to avoid plagiarism.
Page 6 – Font is cut off in blue boxes to left, formatting of figure is needed.
Line 354 – Needs page number for citation
Line 348-350 – Wrong citation format
Interventions – Wrong citation format throughout
Comments on the Quality of English LanguageMinor edits required only
Author Response
Reviewer 1
Comment
Thank you for this opportunity to review this manuscript. It covers an important topic and a scoping review would improve our knowledged on TIC. The intro is very well written, the PRISMA-ScR is covered adequately. Initially my edits were minor. However, when I got to the Interventions, the writing style, citation style, and font style was very inconsistent. I find this concerning for plagairism or AI. I stopped my review at this point based on my concerns. I made the following notations up to this point:
Response
We do not know why the font style was inconsistent. It might have happened doing the upload. The paper was edited by the Health Care, recommended proofread firm, from the MDPI author service. However, we can see that our original manuscript had the writing style Times new Roman which later was changed to Palatino lino. Something must have happened in the transformation from submitting to the review. Also the differing citations style, might have been do to problems with reference manager system zotero. However, we have changed the citations completely using endnote. Moreover, we have used the advised writing style, Palatino lino, throughout the manuscript and hope this continues through the upload.
The comment about plagiarism came as a surprise to us. We have run a check for plagiarism ourselves, and the only place in the text where we find overlap is around the lines 145 – 150
“This scoping review was registered within the Open Science Framework (registration number xxx). We chose a scoping review methodology (Peters et al., 2022), as this type of review is appropriate for synthesizing diverse types of evidence that underpin complex interventions such as TIC. We followed the PRISMA-ScR (Tricco et al., 2018) reporting guideline procedure, which includes 5 steps: (1) formulating the research question(s); (2) identifying relevant studies; (3) selecting studies; (4) mapping data; and (5) collating, summarizing, and reporting the results. We involved stakeholders in different steps of the study (Pollock et al., 2022)”
which are also part of our online registration of the project in the Open Science Framework (registration number https://doi.org/10.17605/OSF.IO/RZSKQ) with the last update on the 20th of December 2023. This might have caused the concern. Thank you for this awareness.
We have rephrased this part of the paper consequently.
We do not know, if the proofread firm used AI, however, we did not use any AI technology. Reviewing the literature and writing this paper has been a long and comprehensive work, lasting for more than 3 years. We divided tasks between us, and all of the authors have written parts of the ms. In order for us to ensure a consequent reporting style, we used the proofread firm, so that it would fit the audience of the HealthCare better. We guarantee that all the writing is original.
We are pleased that reviewer 1 found the manuscript to cover an important topic.
Comment
Line 55 – Would make two sentences. New sentence starting with, “For example, if they…”
Line 70 – I did not see quotes to suppor the “(p. 5)”
Line 145-149 – The text is different in these lines which is concerning for plagiarism. Quotations must be used for direct quotes to avoid plagiarism.
Page 6 – Font is cut off in blue boxes to left, formatting of figure is needed.
Line 354 – Needs page number for citation
Line 348-350 – Wrong citation format
Interventions – Wrong citation format throughout
Response
We have changed the ms to accommodate the remarks above, some changes happened as a consequence of changing the citation style.
Most of the above was corrected when we used a new citation system (End note) and changed the citation style
We rephrased the method section lines 145 – 149 slightly so that the text is not identical to the text in the Open Science Framework
Thank you for the awareness of the Prisma flow chart, we have now refined this.
Reviewer 2 Report
Comments and Suggestions for Authors
Dear Authors
Many thanks for submitting this manuscript for review. Please see below some pointers that can hopefully strengthen the paper further.
Keywords - if of equal importance place in alphabetical order.
Referencing. - Your references should follow Vancouver style throughout as per MDPI guidelines. So instead of Kessler et al., 2017, this should be [1]. Please follow this throughout the text.
No need to add page number to in text citation unless it is a quote.
Considering the topic you are writing on, I would steer away from terms like patient or service user and instead use those who utilise services.
Refrain from using e.g, i.e, etc in academic writing.
Pg.3, line 135 - our objective should be our aim. In addition to the aim, as this is a scoping review you need to document a number of objectives for the paper. For example, see Norton, (2022) protocol - https://bmjopen.bmj.com/content/bmjopen/12/5/e058428.full.pdf
Method - you mention a protocol being registered for the review and provide the link, please also provide the date of registration to the repository.
Formulating the research question - why did you not use an instrument like PICO or SPIDER to create the research question?
The PRISMA flow diagram should be in the results section, not the methods.
The findings are described fine, but became monotonous in places - please consider if such information could be documented in different ways - figures, graphs, more tables and so on.
Once again, thank you for sending this manuscript in for review. I feel with a little more work, the paper would be acceptable for publication in the journal.
Author Response
Reviewer 2
Comment
Many thanks for submitting this manuscript for review. Please see below some pointers that can hopefully strengthen the paper further.
Comment
Keywords - if of equal importance place in alphabetical order.
Response
Thank you for advising, this is corrected
Comment
Referencing. - Your references should follow Vancouver style throughout as per MDPI guidelines. So instead of Kessler et al., 2017, this should be [1]. Please follow this throughout the text.
Response
This has now been corrected, throughout the manuscript. Thank you for this awareness
Comment
No need to add page number to in text citation unless it is a quote.
Response
Thank you, we have now removed unnecessary page numbers.
Comment
Considering the topic you are writing on, I would steer away from terms like patient or service user and instead use those who utilise services.
Response
This is a very good suggestion. We have changed this where it was possible, but sometimes we had to differentiate between organization, service provider and service user, but we totally agree.
Comment
Refrain from using e.g, i.e, etc in academic writing.
Response
We were not aware of this, the paper have been through an academic proof read, but we think your suggestion is good. We have changed it many places, but due to the many citations we have decided to only exemplify some citations in the result section, and therefore we have kept a few e.g. when necessary and changed i.e. and etc. to for example and so on.
But, we agree, we used too many.
Comment
Pg.3, line 135 - our objective should be our aim. In addition to the aim, as this is a scoping review you need to document a number of objectives for the paper. For example, see Norton, (2022) protocol - https://bmjopen.bmj.com/content/bmjopen/12/5/e058428.full.pdf
Response
We have rewritten the objective and was inspired by the suggested NORTON (2002). Thank you for this.
Comment
Method - you mention a protocol being registered for the review and provide the link, please also provide the date of registration to the repository.
Response
This has now been done, with the last updating day.
Comment
Formulating the research question - why did you not use an instrument like PICO or SPIDER to create the research question?
Response
The formulation of the research questions was inspired by Peters et al (2022) suggestions for a scoping review: Population, Concept, and Context and a brainstorming session conducted together with the stakeholders. This was not clear in our reporting and therefore we have amended this. The process was iterative, going forth and back, we hope this is well explained now.
Comment
The PRISMA flow diagram should be in the results section, not the methods.
Response
We have not heard this before? We think it is appropriate to have this in the method section, so that readers can follow the flow.
Comment
The findings are described fine, but became monotonous in places - please consider if such information could be documented in different ways - figures, graphs, more tables and so on.
Response
The material is enormous, we have tried to change the wording here and there. We think we have a good mix of figures and descriptions as is. If it is still a concern for the reviewer, it would be helpful with specific examples, so we know which paragraphs are meant. It was not mentioned by the other 3 reviewers, but we have tried to do it less monotonous.
Comment
Once again, thank you for sending this manuscript in for review. I feel with a little more work, the paper would be acceptable for publication in the journal.
Response
Thank you for your helpful review.
Reviewer 3 Report
Comments and Suggestions for Authors
Thank you for the opportunity to review this paper. There is such a volume of research in this field, a synthesis is regularly needed to build consensus. The authors have done a tremendous job of exploring and synthesizing the research which exists in TIC in a novel manner. The use of the publication graph is extremely interesting and novel, and the use of a stakeholder committee to inform the review is especially valuable.
With this in mind, I would recommend the authors add a section that describes the formation of the stakeholder committee and outlines their roles and responsibilities and any ethics approvals (if any) required to involve them in the contribution. This is a novel and valuable approach that could be made more apparent.
This was a very well considered, timely and comprehensive analysis of TIC in the existing literature.
Author Response
Reviewer 3
Comment
Thank you for the opportunity to review this paper. There is such a volume of research in this field, a synthesis is regularly needed to build consensus. The authors have done a tremendous job of exploring and synthesizing the research which exists in TIC in a novel manner. The use of the publication graph is extremely interesting and novel, and the use of a stakeholder committee to inform the review is especially valuable.
With this in mind, I would recommend the authors add a section that describes the formation of the stakeholder committee and outlines their roles and responsibilities and any ethics approvals (if any) required to involve them in the contribution. This is a novel and valuable approach that could be made more apparent.
Response
This is a very timely question. We have added more to the description of knowledge users in the method section and in the discussion. Moreover, have we added ethical consideration in the finale statement.
Line: 152 – 161, 177 – 183, 624 – 632,
Comment
This was a very well considered, timely and comprehensive analysis of TIC in the existing literature.
Response
Thank you
Reviewer 4 Report
Comments and Suggestions for Authors
Overall review
Brief sumary:
I am glad I got to read this script. It is urgent research as traumatic experiences can have long-term negative effects on not only individuals but also on organizations and society. Trauma-informed care (TIC) could play a central role in mitigating these negative consequences. But since the authors found it is not known how and in what way TIC should be implemented, this review was carried out. The objective was to conduct a scoping review that explores and systematically maps the research conducted in this area and identifies existing gaps in knowledge. The overall research question was "What is known about implementing TIC?” Studies that have implemented TIC were included across human service settings. However, the authors state that they were specifically interested in mental healthcare.
Specific comments:
In a paragraph beginning on line 106, you described early reviews, the most recent of which was published as late as 2023. Try to explain why these reviews could not be compiled and answer your purpose rather than conducting your scoping review.
Rational: You write just before the purpose that a broad overview of how TIC has been implemented across settings and populations is important to establish the current state of knowledge and to guide future decisions about implementing TIC in organizations. The title of your script is not entirely clear in relation to this statement. The title certainly states that it will be about implementing trauma-informed care, but the reader is also informed that it will be about definitions, interventions and measures, not implementing TIC across settings. Consider clarifying the title to match your rationale.
Aim: Explain why you didn't just stick to describing implementing TIC across settings but to put a specifically interested in mental healthcare. Could it be that the latter would have been better carried out with the help of one a systematic overview that tries to compile empirical evidence from a relatively smaller number of studies that relate to a focused research question?
Areas of the strength of this paper
Introduction:
The chosen research problem is problematized, motivated and defined based on the current state of the art.
The introduction is predominantly clear and related to the problem area.
Materials and Methods
· The choice of method is described and it is clear what the selection is. The approach in your scoping review is well described and gives a credible impression with suitable databases and an updated survey in 2022.
· I see it as strength to involve stakeholders, to ensure that the aims of the study reflects their needs.
· Data analysis is well described and credible.
Results
· The results is structured and based on a correct and systematic analysis. It gives the results section a clarity that you divided it into the three subsections: an overview of the characteristics of the included studies and their methodologies; the summarize of key concepts and definitions used across studies and the description of the interventions and measures used.
· The tables and figure are helpful.
Discussion
· Main results are discussed with a reflective and critical approach in relation to the current state of research. Strengths and weaknesses in relation to the result are discussed with a critical approach.
· It is an interesting and informed piece of discussion that begins on line 501 about, among other things, your use of stakeholder. It is a complicated and sometimes difficult process to create a review, which you demonstrate.
I am pleased to read the text in the paragraph from lines 628 to 636 where you write about a need for more simplistic and unanimous approaches to implementing TIC where steps or phases of implementation are more clearly delineated. I agree that TIC initiatives should be informed by implementation science, where the organizational context is taken into account. Implementing TIC really means complex intervention and an MRC, Medical Research Council, framework dealing with developing and evaluating complex interventions could be helpful, as I see it. Much has been written about the Medical Research Council framework, but I see the two books below as original, after which a number of publications have been written:
Richards, D. A., & Hallberg, I. R. (Eds.). (2015). Complex interventions in health: an overview of research methods.
Richards, D. A., & Hallberg, I. R. (2015). Implementation of complex interventions. In Complex Interventions in Health (pp. 259-272). Routledge.
You mention Skivington, et al (2021) A new framework for developing and evaluating complex interventions: Update of Medical Research Council guidance. BMJ, reference nr 78. But perhaps you could have developed this a little more in the discussion section.
Areas of the weakness of this paper
Introduction
· You write in the last paragraph before Aims and research questionst that to your the best knowledge, there have been no published reviews that provide abroad overview of studies across healthcare settings and other human services.
But since you partly write that you want to contribute with how TIC has been implemented across settings but also that you focused on mental health in particular, the following question need a clarification:
Why is it not enough with the reviews that already exist or
could not these reviews be compiled and answer your purpose rather than conducting your scoping review?
Materials and Methods
· The idea to involve stakeholders, to ensure that the obstacle reflects their needs is interesting and could be of great use. But it seems difficult to find suitable, sufficiently numerous and diverse stakeholders to be able to reflect such a far-reaching research question as "What is known about implementing TIC?" You need to explain and clarify this.
Results
· I believe that you carried out a good study, that it turned out to be an interesting and valuable result, despite the limitations I mentioned. You also write yourself that there are several limitations of this review as it was an enormous undertaking.
Discusssion
· In the first paragraph of the discussion, lines 495-497, you write that your review provides a more comprehensive overview of the literature, as you did not limit your search to specific settings or populations. However you declare in the method section that you that you focused on mental health in particular, where you refer to the literature search. I am confused by this ambiguity. It would require clarification.
References
The references are mostly very relevant and recent. A few references are old, e.g. no. 8,16, and 20, but I can understand that you include them to do justice to the diversities in the study.
Author Response
Brief sumary:
Comment
I am glad I got to read this script. It is urgent research as traumatic experiences can have long-term negative effects on not only individuals but also on organizations and society. Trauma-informed care (TIC) could play a central role in mitigating these negative consequences. But since the authors found it is not known how and in what way TIC should be implemented, this review was carried out. The objective was to conduct a scoping review that explores and systematically maps the research conducted in this area and identifies existing gaps in knowledge. The overall research question was "What is known about implementing TIC?” Studies that have implemented TIC were included across human service settings. However, the authors state that they were specifically interested in mental healthcare.
Response
Yes, we have rewritten the objective and tried to clarify in the paper why we aimed to include mental health care/ psychiatry. Without having this limitation the amount of papers was to big (as you can see in table 1 example of search strategy). Mental Health goes across organizations, settings and society. It is preferable to implement TIC across settings, however as our stakeholders was working within mental healthcare setting we chose this, but as we also describe, this is an iterative process, and during the process, we became more aware of the different contexts. Even though we included words such as "psychiatry" in our search string, most of the 3000+ studies that were identified were conducted in other settings e.g. child welfare settings. We decided to focus broadly on all the varied settings in the studies as this reflected our data and is aligned with the concept of a scoping review. We have added some reflections about this in the method section line 140 – 143, and L. 155 – 162, and in the discussion line 636 - 638
Specific comments:
In a paragraph beginning on line 106, you described early reviews, the most recent of which was published as late as 2023. Try to explain why these reviews could not be compiled and answer your purpose rather than conducting your scoping review.
Response
When we began the review, we were not aware of these reviews. However, these reviews interpret, minor studies and gives a narrow picture of the field. We wanted to make a broader scope and include end-users perspectives and wishes. We have explained this further in the introduction just before the aim around Line 119 – 127, and added a clarification in Line 224. Furthermore, none of the previous reviews provide an overview of existing reviews, which we do. However, our aim which was defined partly by end-users was not to review existing reviews. We have also reflected on this in the limitations L. 688 - 692
Comment
Rational: You write just before the purpose that a broad overview of how TIC has been implemented across settings and populations is important to establish the current state of knowledge and to guide future decisions about implementing TIC in organizations. The title of your script is not entirely clear in relation to this statement. The title certainly states that it will be about implementing trauma-informed care, but the reader is also informed that it will be about definitions, interventions and measures, not implementing TIC across settings. Consider clarifying the title to match your rationale.
Response
Thank you for this very good remark. We have renamed the titel to: Implementing Trauma-Informed Care—Settings, Definitions, Interventions, Measures and Implementation Across Settings: A Scoping Review"
Comment
Aim: Explain why you didn't just stick to describing implementing TIC across settings but to put a specifically interested in mental healthcare. Could it be that the latter would have been better carried out with the help of one a systematic overview that tries to compile empirical evidence from a relatively smaller number of studies that relate to a focused research question?
Response
We involved gatekeepers and they helped us to become aware of some of the challenges of doing a narrow review. Also, it was necessary with this broad scope to realize challenges and opportunities. Such as challenges in how to measure long term impact combined with challenges in relation to different interventions. Our review shows a huge complexity, which is not apparent if we had for example just focused on either implementation or interventions. When reading about the TIC approach individuals become enthusiastic, however, diving into the complex reality makes it much more difficult to implement TIC, and our review illustrates why, which could help us move forward, as we have sought to capture this in the discussion.
Comment
Areas of the strength of this paper
Introduction:
The chosen research problem is problematized, motivated and defined based on the current state of the art. The introduction is predominantly clear and related to the problem area.
Materials and Methods
- The choice of method is described and it is clear what the selection is. The approach in your scoping review is well described and gives a credible impression with suitable databases and an updated survey in 2022.
- I see it as strength to involve stakeholders, to ensure that the aims of the study reflects their needs.
- Data analysis is well described and credible.
Results
- The results is structured and based on a correct and systematic analysis. It gives the results section a clarity that you divided it into the three subsections: an overview of the characteristics of the included studies and their methodologies; the summarize of key concepts and definitions used across studies and the description of the interventions and measures used.
- The tables and figure are helpful.
Discussion
- Main results are discussed with a reflective and critical approach in relation to the current state of research. Strengths and weaknesses in relation to the result are discussed with a critical approach.
- It is an interesting and informed piece of discussion that begins on line 501 about, among other things, your use of stakeholder. It is a complicated and sometimes difficult process to create a review, which you demonstrate.
I am pleased to read the text in the paragraph from lines 628 to 636 where you write about a need for more simplistic and unanimous approaches to implementing TIC where steps or phases of implementation are more clearly delineated.
Response
Thank you for this feedback
Comment
I agree that TIC initiatives should be informed by implementation science, where the organizational context is taken into account. Implementing TIC really means complex intervention and an MRC, Medical Research Council, framework dealing with developing and evaluating complex interventions could be helpful, as I see it. Much has been written about the Medical Research Council framework, but I see the two books below as original, after which a number of publications have been written:
Richards, D. A., & Hallberg, I. R. (Eds.). (2015). Complex interventions in health: an overview of research methods.
Richards, D. A., & Hallberg, I. R. (2015). Implementation of complex interventions. In Complex Interventions in Health (pp. 259-272). Routledge.
Response:
Yes, we are aware of these books, and use them a lot. We can change this if you want or add on. However, there are already many citations and we decided on the review of Nissen (Citation 210) as they explain the importance of context, as is also relevant in relation to TIC.
Comment
You mention Skivington, et al (2021) A new framework for developing and evaluating complex interventions: Update of Medical Research Council guidance. BMJ, reference nr 78. But perhaps you could have developed this a little more in the discussion section.
Response
This is absolutely a relevant remark - this was too superficial. We have elaborated on this in Line 690 - 697
Comment
Areas of the weakness of this paper
Introduction
- You write in the last paragraph before Aims and research questionstthat to your the best knowledge, there have been no published reviews that provide abroad overview of studies across healthcare settings and other human services.
But since you partly write that you want to contribute with how TIC has been implemented across settings but also that you focused on mental health in particular, the following question need a clarification:
Why is it not enough with the reviews that already exist or could not these reviews be compiled and answer your purpose rather than conducting your scoping review?
Response
We have answered this question above, however we have tried to clarify why this review is relevant. It is relevant because a focused question will not answer the overall aim of the study.
When we began the review, we were not aware of these reviews. However, these reviews interpret, minor studies. We wanted to make a broader scope and include end-users perspectives and wishes. We have explained this further in the introduction just before the aim around Line 119 – 127 and added some reflections in the discussion Line 638 and in the limitation section L. 688 - 692.
Moreover, it would not have showed the rise in figure 2
It could be interesting to do a review of reviews, and this might be important to do. But it will not open up for the complexities we have identified.
Comment
Materials and Methods
- The idea to involve stakeholders, to ensure that the obstacle reflects their needs is interesting and could be of great use. But it seems difficult to find suitable, sufficiently numerous and diverse stakeholders to be able to reflect such a far-reaching research question as "What is known about implementing TIC?" You need to explain and clarify this.
Response
We have described the stakeholders more in depth, such as explaining how they were engaged in a network aiming to work towards a TIC approach in Denmark, see line: 146 - 165
Comment
Results
- I believe that you carried out a good study, that it turned out to be an interesting and valuable result, despite the limitations I mentioned. You also write yourself that there are several limitations of this review as it was an enormous undertaking.
Comment
Yes, and thank you
Discusssion
- In the first paragraph of the discussion, lines 495-497, you write that your review provides a more comprehensive overview of the literature, as you did not limit your search to specific settings or populations. However you declare in the method section that you that you focused on mental health in particular, where you refer to the literature search. I am confused by this ambiguity. It would require clarification.
Response
We understand this issue, and we have tried to clarify this several places amongst others Line 641 – 645, as well as in the method section line 160, and in the search strategy.
Comment
References
The references are mostly very relevant and recent. A few references are old, e.g. no. 8,16, and 20, but I can understand that you include them to do justice to the diversities in the study.
Response
Yes, Tic is developed through many years, which also is reflected in figure 2, with the rise in publications. It is important to understand the history in order to guide the future. So we hope this is ok.
Round 2
Reviewer 4 Report
Comments and Suggestions for Authors
I believe that you carried out a good work with the revision and completion, that it turned out to be an even more interesting and valuable manuscript. After the manuscript has been revised and supplemented, I consider it ready for publication. You have answered my criticism well and the manuscript has been well reworked. It will become a valuable contribution to the research world, useful the health and medical care that wish to implement the TIC concept.